# Risk Reclassification of Patients with Endometrial Cancer Based on Tumor Molecular Profiling: First Real World Data

**DOI:** 10.3390/jpm11010048

**Published:** 2021-01-15

**Authors:** Felicitas Oberndorfer, Sarah Moling, Leonie Annika Hagelkruys, Christoph Grimm, Stephan Polterauer, Alina Sturdza, Stefanie Aust, Alexander Reinthaller, Leonhard Müllauer, Richard Schwameis

**Affiliations:** 1Department of Pathology, Medical University of Vienna, 1090 Vienna, Austria; felicitas.oberndorfer@meduniwien.ac.at (F.O.); n1542004@students.meduniwien.ac.at (L.A.H.); leonhard.muellauer@meduniwien.ac.at (L.M.); 2Comprehensive Cancer Center, Gynecologic Cancer Unit, Department of Obstetrics and Gynecology, Medical University of Vienna, 1090 Vienna, Austria; sarah.moling@gmail.com (S.M.); stephan.polterauer@meduniwien.ac.at (S.P.); stefanie.aust@meduniwien.ac.at (S.A.); alexander.reinthaller@meduniwien.ac.at (A.R.); richard.schwameis@meduniwien.ac.at (R.S.); 3Karl Landsteiner Institute for General Gynecology and Experimental Gynecologic Oncology, 1090 Vienna, Austria; 4Comprehensive Cancer Center, Gynecologic Cancer Unit, Department of Radiation Oncology, Medical University of Vienna, 1090 Vienna, Austria; alina.sturdza@akhwien.at

**Keywords:** endometrial cancer, cancer gene panel sequencing, molecular tumor classification

## Abstract

Recently, guidelines for endometrial cancer (EC) were released that guide treatment decisions according to the tumors’ molecular profiles. To date, no real-world data regarding the clinical feasibility of molecular profiling have been released. This retrospective, monocentric study investigated the clinical feasibility of molecular profiling and its potential impact on treatment decisions. Tumor specimens underwent molecular profiling (testing for genetic alterations, (immune-)histological examination of lymphovascular space invasion (LVSI), and L1CAM) as part of the clinical routine and were classified according to the European Society for Medical Oncology (ESMO) classification system and to an integrated molecular risk stratification. Shifts between risk groups and potential treatment alterations are described. A total of 60 cases were included, of which twelve were excluded (20%), and eight of the remaining 48 were not characterized (drop-out rate of 16.7%). Molecular profiling revealed 4, 6, 25, and 5 patients with DNA polymerase-epsilon mutation, microsatellite instability, no specific molecular profile, and TP53 mutation, respectively. Three patients had substantial LVSI, and four patients showed high L1CAM expression. Molecular profiling took a median of 18.5 days. Substantial shifts occurred between the classification systems: four patients were upstaged, and 19 patients were downstaged. Molecular profiling of EC specimens is feasible in a daily routine, and new risk classification systems will change treatment decisions substantially.

## 1. Introduction

Endometrial cancer (EC) is the most frequent cancer of the female genital tract, and in the Western world, one in 35 women will develop EC during their lifetime [1]. Historically and based on an observational trial, EC was classified into two subtypes (type 1/type 2) [2]. In 2004, the World Health Organization (WHO) classified EC based on histopathologic parameters as endometrioid, serous, clear cell, or carcinosarcoma. 

While the WHO classification is well established and pragmatic, there are substantial problems in reproducibility [3]. Currently, recommendations for adjuvant therapy after surgery are based upon the WHO classification system and additional clinicopathologic parameters. Traditional clinicopathologic parameters include tumor invasion depth, histologic grading, patient age, and lymph vascular space invasion (LVSI). 

Recently, large prospective trials such as the “post-operative radiotherapy for endometrial cancer” (PORTEC) studies have aided the adaptation of adjuvant therapy in patients with EC [4,5]. These studies have helped to minimize overtreatment while achieving high quality of life and improved clinical outcome for patients. However, current analyses show that despite all treatment efforts, 8% of patients with high–intermediate risk EC will develop distant metastases, and that currently 7 patients are treated with vaginal brachytherapy to prevent one case of local recurrence [6,7]. 

In 2014, the comprehensive genomic profiling of the Cancer Genome Atlas (TGCA) classified four molecular types of EC [8]. These four groups consist of tumors with mutations in the exonuclease domain of DNA polymerase-epsilon (POLE), tumors with mismatch repair deficiency (microsatellite instable group, MSI), EC with a high gene copy number alteration which is mostly driven by TP53 mutation, and a group of cancers with low gene copy number changes (also referred to as ‘no specific molecular profile’, NSMP). The prognostic value of the four molecular subgroups has been shown in several cohorts [9,10]. Meanwhile, additional molecular pathologic risk factors have been described, including neural cell adhesion molecule L1 (L1CAM) expression, a new nomenclature of LVSI, and mutations in the CTNNB1 gene [11,12,13]. One study investigated the PORTEC-1 and -2 cohorts, performed a hotspot mutation analysis of 14 genes, classified all patients according to the four molecular subtypes, and in a further step, performed analysis of L1CAM, LVSI, and CTNNB1 [12,14]. The integration of these risk factors allowed the stratification of EC into a novel integrated molecular risk classification including three risk groups and corroborated the prognostic relevance [14]. Hence, treatment recommendations based on these molecular risk factors have been established, which are currently under evaluation in an ongoing prospective trial (PORTEC-4a, NCT03469674). In addition, the prognostic value of the molecular subgroups in high-risk EC has been validated in a retrospective analysis of the PORTEC-3 cohort [15,16] and a treatment algorithm has been proposed [17]. 

However, whereas all of the previously mentioned studies demonstrated the prognostic value of molecular profiling and the integrated risk stratification of EC, no data have hitherto been published regarding the feasibility of classifying EC according to the integrated molecular pathological system in a clinical setting. This study aimed at evaluating over a one-year period the feasibility of postoperative analysis of EC specimens according to the integrated genomic classification system in a university clinical setting of a high-volume gynecologic oncologic center in Central Europe. Additionally, it was evaluated to what extent assignment to prognostic subgroups and treatment would have differed if the patients had been treated as currently investigated in prospective trials.

## 2. Materials and Methods

We performed a retrospective analysis of all patients undergoing treatment for primary endometrioid EC at the Comprehensive Cancer Center, Medical University of Vienna, Austria, between October 2017 and October 2018. Patients’ clinical data were gathered from patients’ electronic medical records. Only patients receiving their complete treatment at our institution were included, while patients with nonendometrioid histology findings in their final reports were excluded from the study. 

### 2.1. Patients

Diagnosis of EC was established histologically with tissues obtained either by dilation and curettage or endometrial biopsy. Clinical assessment including transvaginal ultrasound was performed by a specialist in gynecologic oncology. In addition, computed tomography (CT) and magnetic resonance imaging (MRI) were preoperatively performed to assess tumor stage. 

Patients were treated according to local standards and international guidelines [18]. In stage I–III disease, hysterectomy, bilateral salpingo oophorectomy, and sentinel lymphadenectomy using indocyanine green were recommended. Furthermore, if bulky disease or peritoneal carcinosis was detected during surgery, debulking surgery was performed. Adjuvant treatment was administered according to current ESMO-ESGO-ESTRO (European Society for Medical Oncology, European Society of Gynaecological Oncology, European Society for Radiotherapy and Oncology) guidelines [18]. All patients were included in the institution’s follow-up program that encompasses clinical examination and imaging for a total of 10 years, when necessary. 

### 2.2. Ethics

This study was approved by the Institutional Review Board of the Medical University of Vienna (IRB approval number: 2014/2019). All patients gave their consent to treatment according to institutional guidelines and to anonymized assessment of clinical data and treatment outcome. In accordance with Austrian law, all patients gave specific written informed consent to molecular and genetic analysis of their EC specimens. Since the current study represents a retrospective analysis, the Institutional Review Board of the Medical University of Vienna waived the requirement to obtain distinct written informed consent from each patient. Patient data were anonymized and de-identified prior to analysis. The study was performed according to the Declaration of Helsinki, the ICH Harmonized Tripartite Guideline for Good Clinical Practice, and the guidelines of the Institutional Review Board of the Medical University of Vienna.

### 2.3. Pathologic Processing

Resection specimens obtained during surgery were processed and diagnosed according to the 4th edition of the WHO Classification of Tumors of Female Reproductive Organs [19]. Pathological examinations and reports were conducted by experienced gynecopathologists as part of their routine work. Tumor stage and grade were determined according to the UICC TNM classification of malignant tumors, 8th edition. In this study, we included only patients at tumor stage I–III. The tumor stage and grade were determined at the time of rendering the pathology report for tumor stage as pT, pN, and for grade as G1, G2, or G3, according to the International Federation of Gynecology and Obestrics (FIGO) grading scheme [20]. To exclude metastatic disease, a CT scan was performed. Next-generation sequencing (NGS) of a 161 cancer gene panel and the DNA polymerase genes POLD1 and POLE, and DNA mismatch repair protein expression analysis by immunohistochemistry and microsatellite instability (MSI) testing by microsatellite length determination, were conducted with formalin-fixed, paraffin-embedded (FFPE) tissue concurrently at the time of diagnosis. Neural cell adhesion molecule L1 (L1CAM) immunohistochemistry was performed on a representatively chosen tissue slide retrospectively and assessed by the authoring pathologists (F.O., L.M.) blinded to the histopathological report and clinical data.

### 2.4. Cancer Gene Panel Sequencing

For DNA extraction, FFPE tissue was macro-dissected to increase the tumor cell content by employing a corresponding hematoxylin-and-eosin-stained tissue slide with a marked tumor area. DNA was extracted from FFPE tissue blocks using an EZ1^®^ DNA Tissue Kit (48) (Quiagen, Hilden, Germany) and treated with uracil-DNA glycosylase (UDG) to reduce formalin artefacts that might reduce the accuracy of sequencing [21]. For DNA quantification, a Qubit 2.0 fluorometer was used. The DNA libraries for sequencing were constructed with 10 ng DNA per library for multiplex polymerase chain reactions (PCR) with the OncomineTM Comprehensive Assay v3 (Thermo Fisher Scientific, Waltham, MA, USA) and a custom ampliseq panel (Thermo Fisher) comprising all exons of the POLE and POLD1 genes. The OncomineTM Comprehensive Assay v3 includes 161 cancer genes, covering mutation hotspots in 87 genes, all coding exons of additional 48 genes, and copy number gains in 43 genes. DNA was sequenced with the Ion GeneStudio™ S5 System (Thermo Fisher) and the sequences were analyzed with Variant Caller TM and Ion Reporter TM (both from Thermo Fisher) software and an in-house developed bioinformatics annotation tool. Tumor purity was defined as percentage of tumor cells in the tissue or encircled area of the tissue. A mutation was classified as subclonal if the variant allele frequency (VAF) of a mutated allele was ≤10% of the estimated tumor cell content [22]. 

### 2.5. Immunohistochemistry and Microsatellite Instability Testing

The expression of the four DNA mismatch repair (MMR) proteins MLH1, MSH2, MSH6, and PMS2 and the cell adhesion molecule L1CAM was determined by immunohistochemistry with one representative FFPE tissue block for each tumor. The MMR protein expression was diagnosed at the time of molecular pathological workup, and L1CAM staining was performed after the study population was definite. Immunostaining was performed on 2 µm thin tissue sections with a Benchmark Ultra autostainer (Ventana Medical Systems, Tucson, AZ, USA) [23,24]. The antibodies used are listed in Table 1. Immunostainings were carried out following in-house validated protocols. Briefly, slides containing unstained sections were pretreated performing heat-induced epitope retrieval in cell conditioning 1 (CC1) buffer (heating temperature 95 °C), pH 8 (Ventana Medical Systems) for 52 min (MLH1 and MSH2), 64 min (MSH6), 92 min (PMS2), and 36 min (L1CAM). Incubation time for primary antibody was 40 min (MLH1), 16 min (MSH2), 32 min (MSH6), 1 h 20 min (PMS2), and 32 min (L1CAM) at 36 °C, respectively. For visualization of antigen–antibody binding, the ultraView Universal DAB detection kit (Ventana/Roche) was used, which contains a cocktail of horseradish peroxidase (HRP)-labeled antibodies (goat anti-mouse IgG, goat anti-mouse IgM, goat anti-rabbit). To preclude unspecific binding of antibodies, we performed negative controls by omitting the primary antibodies and by employing isotype-specific control antibodies at the same concentrations and laboratory conditions as the primary antibodies. For MLH1, MSH2, MSH6, and L1CAM stainings, we used Negative Control (Monoclonal) # 760-2014, and for PMS2, CONFIRM™ Negative Control Rabbit Ig, # 760-1029 (both from Ventana Medical Systems).

The immunohistochemical stainings of MMR proteins were diagnosed by either one of the authoring pathologists (F.O. or L.M.) as part of diagnostic workup. A retained nuclear immunoreactivity with antibodies to MLH-1, MSH-2, MSH6, and PMS2 of invasive carcinoma cells and normal tissue cells was regarded as DNA MMR proficient and therefore having a very low likelihood of microsatellite instability. It was considered normal if expression was patchy or faint [25]. Tumors with no nuclear reactivity for at least one of the four MMR proteins and ambiguous cases were interrogated for MSI with the MSI analysis system version 1.1. from Promega (Madison, WI, USA) [26]. This system comprises five nearly monomorphic mononucleotide microsatellites (MONO-27, NR-21, NR-24, BAT-25, and BAT-26) and two highly polymorphic pentanucleotide markers (Penta C and Penta D) that are amplified by PCR from matched carcinoma and normal DNA. Amplified microsatellites were detected and their length distribution compared by capillary electrophoresis on a 3500 Genetic Analyzer (Applied Biosystems, Waltham, MA, USA).

The immunohistochemical staining of L1CAM was interpreted by the authoring pathologists (F.O. and L.M.) blinded to clinical and pathological reports. Discrepant cases were discussed until a consensus interpretation was reached. The expression of L1CAM was quantified by determination of the percentage of carcinoma cells with L1CAM immunoreactivity. Additionally, the intensity of the staining was graded as low, medium, or dense, as described by Zeimet et al. [12]. Furthermore, LVSI was scored as focal or substantial as previously described [11].

Representative immunohistochemical images of MMR proteins and L1CAM are provided as Appendix A.

### 2.6. Risk Group Classification

All patients were classified (1) into risk groups to guide adjuvant therapy as recommended by ESMO-ESGO-ESTRO guidelines [18] and (2) according to the ECs molecular profile as defined by the TCGA [8]. In addition, patients with low to high intermediate risk disease were classified into the risk groups defined by Stelloo et al. [14] using additional molecular pathologic parameters including CTNNB1 mutation status and L1CAM and LVSI status as performed in the PORTEC-4a study. Patients with high risk disease—following the definition used in the PORTEC-3 cohort [15,16]—were classified into four risk groups as suggested by the recently published ESGO/ESTRO/ESP guidelines [17,27]. 

### 2.7. Statistical Analysis

All data are given either as the mean (SD) or median (range). Patients were classified according to the 2009 International Federation of Gynecology and Obstetrics (FIGO) classification system [28] and the 4th edition of the WHO Classification of Tumors of Female Reproductive Organs [19]. Chi-squared tests were performed to compare the stratification systems. A Sankey diagram was generated to illustrate shifts of patients between risk groups of different classification systems using power-user add-in for MS Excel (Power User Software, Paris, France). *p*-values of 0.05 were considered to be statistically significant. Statistical analysis was performed using the Statistical Package for the Social Sciences statistical software (SPSS 24.0 for MAC, IBMCorp., Armonk, NY, USA).

## 3. Results

In total, 60 EC patients were treated between October 2017 and October 2018 at the Medical University of Vienna, Austria. Twelve patients had to be excluded from the study: (i) two patients had surgery in another cancer center and solely received adjuvant therapy at our department; (ii) the final histology of seven patients revealed non endometrioid EC; (iii) in two patients, imaging studies revealed metastatic disease; (iv) in one patient, only hormone therapy was administered for fertility-sparing reasons. Finally, 48 patients were eligible for the current analysis. Molecular profiling was recommended in all 48 patients but was only performed in 40 cases. Therefore, the clinical drop-out rate was 16.7%, as 8/48 cases did not undergo molecular profiling due to the respective clinician in charge not honoring the request at the time of surgery. Table 2 shows demographics of the study population. 

Conclusive molecular profiling results were obtained in all 40 cases. All standard pathologic results were available within ten days after surgery, and molecular profiling took a median of an additional 18.5 days. Therefore, the median duration until all results were available (calculated from the day of surgery to the day of final molecular result) was 28.5 (9–75) days.

In total, four tumors were classified as POLE-mutated, six tumors as MSI, five tumors as TP53-mutated, and 25 as NSMP (Figure 1). Detailed information on detected pathogenic/likely pathogenic mutations of potential driver genes according to TCGA data on Uterine Corpus Endometrial Carcinoma (accessed by cBioPortal 27.4.2020, www.cbioportal.org), including descriptions of the mutations at the DNA and protein sequence levels and the respective reference sequences, are given in Appendix A. Patient 5 showed a POLE mutation with a VAF of 3.5%, which was classified as subclonal due to 80% tumor cell content and a VAF lower than 10% of the tumor cell content. Furthermore, the tumor of this patient showed a complete loss of MSH6 staining. Therefore, this patient was classified into the MSI group. Interestingly, the tumors of patients 6, 11, 14, and 31 did not shown any genetic alteration. The tumor of patient 6 showed a complete loss of MLH1 and PMS2 staining and was therefore classified into the MSI group. The tumors of patients 11, 14, and 31 were classified as NSMP. Patient 35 showed a TP53 (exon 8) mutation with a VAF of 7%. It was classified as subclonal due to 90% tumor cell content and a VAF lower than 10% of the tumor cell content. Therefore, this patient was classified as having NSMP.

Twenty-three tumors were classified as LVSI negative, 14 tumors showed mild LVSI, and three tumors had substantial LVSI. In addition, 11 tumors expressed L1CAM, of which four exhibited more than 10% L1CAM-positive cells. Patients were classified according to ESMO risk groups, molecular groups, and integrated molecular risk groups. Table 3 compares the ESMO risk groups with both the TCGA molecular subtypes and the integrated molecular risk groups. 

Integrative molecular profiling led to several shifts within the risk groups. In total, 23 out of 40 patients (57.5%) shifted between risk groups. Upstaging was performed in four patients (10%): two patients were upstaged from the low-risk group to the high-risk group, and another two patients were upstaged from the high–intermediate-risk group to the high-risk group. In contrast, 19 patients (47.5%) were downstaged: three patients from the intermediate-risk group, one patient from the high–intermediate-risk group, and four patients from the high-risk group were downstaged to the low-risk group, in addition to 11 patients of the high-risk group who were downstaged to the high–intermediate-risk group. To illustrate how differently patients are classified by the available classification systems, Figure 2 shows the shifts of patients between the ESMO and integrated molecular risk groups.

## 4. Discussion

This analysis reports on the first real-world data of molecular profiling in women with endometrial cancer. We have demonstrated that the integration of molecular profiling into a daily routine is possible. All patients who underwent molecular profiling could be assigned to one of the risk groups, without any multiple classifiers, i.e., tumors with features of two or more different risk groups. 

In 23 patients, molecular profiling lead to shifts between risk groups. Interestingly, risk group shifts were not limited to patients with FIGO stage 1 disease, but occurred within all FIGO stages: In FIGO stage 1, 4/30 (13.3%) patients were upstaged to a higher risk group, while 11/30 (36.7%) patients were downstaged. In FIGO stages 2 and 3, 6/7 patients (85.7%) and 2/3 (66.7%) patients were downstaged, respectively. 

Notably, risk stratification according to molecular profiling would have changed clinical treatment decisions in twelve patients. Four patients would have received additional treatment including external beam radiotherapy in three cases and combined vaginal brachytherapy and chemotherapy in one case. In eight cases, treatment would have been de-escalated. Specifically, in six cases, vaginal brachytherapy; in one case, external beam radiotherapy; and in one case, combined external beam radiotherapy and chemotherapy would have been spared. 

As indicated by recent data, POLE-hypermutated tumors seem to bear a favorable prognosis compared to the other risk groups, and de-escalation of adjuvant therapy in this population has been suggested [29]. In line with these considerations, data of the PORTEC-3 trial indicated that the combination of chemotherapy with radiotherapy showed no benefit compared to radiotherapy alone in adjuvant treatment of POLE-mutated cancer [16]. In the present cohort, one patient was staged as FIGO stage 3, G3 disease with lymph node involvement. This tumor was classified as POLEmut. According to the ESMO classification system, this patient was classified into the high-risk group, and was therefore treated with external radio beam therapy as well as chemotherapy. Based upon molecular profiling [10], POLEmut tumors are considered low risk irrespective of tumor stage, and therefore no adjuvant therapy would have been recommended to this patient. 

Six patients had MSI tumors that tend to express a very high number of tumor neo-antigens, leading to high response rates for immunotherapy [30,31]. While none of the patients included into this study received immunotherapy, the use of immunotherapy in MSI EC is comprehensively described and the subject of several ongoing prospective trials [30]. In addition, the current ESGO guideline recommends to consider immunotherapy with pembrolizumab as second-line therapy of MSI tumors [27].

The tumors of five patients in this trial featured a TP53 mutation. Three of these patients were subjected to adjuvant chemotherapy, while the other two patients received no adjuvant therapy. However, based on recent data [32], future prospective studies should investigate adjuvant chemotherapy or chemoradiotherapy for all patients with TP53 mutated tumors or p53 overexpression on IHC, independent of tumor stage and conventional risk stratification [16]. 

The majority of tumors (25/40) were classified as NSMP because they showed no MSI and no POLE or TP53 mutation. The tumors of patients 11, 14, and 31 did not even show any genetic alteration. The fact that the majority of the tumors were classified as NSMP underlines the importance of further classification of tumor by use of additional methods such as L1CAM expression and LVSI analysis. In the current analysis, these additional methods could identify two of 25 patients with high L1CAM expression, and hence these tumors were classified as high-risk tumors. One of the major goals in the near future must be to find further risk factors that help to reclassify these tumors and to narrow down the NSMP group. 

Molecular profiling seems feasible in a daily routine, however not without effort. One has to consider the additional time required for molecular workup, which may delay adjuvant therapy. In our facility, molecular profiling took a median of 18.5 days in addition to ten days histopathologic workup. Therefore, pathologic analysis took roughly 4 weeks in total. Adjuvant therapy should be initiated within eight weeks after surgery [33] in order to avoid early progression [34]. Hence, decision making based on molecular profiling leaves a time margin of roughly 4 weeks to plan adjuvant treatment appointments. However, in six out of 40 cases (15.0%), analysis took over six weeks, thus leaving only two weeks for the start of adjuvant therapy, which might leave clinicians with only a very narrow time margin to initiate additional therapy. This delay could, of course, be avoided by reflex testing, which is currently on the verge of implementation. This means that every endometrial carcinoma is subjected to molecular profiling irrespective of a clinical request. 

Another upcoming topic which is gaining broader influence in diagnostics is digitalization in pathology. As digital pathology progresses with providing whole slide images, image analysis software is already used, e.g., to analyze immunohistochemistry, although to date, it has not been widely used in daily practice. By implementing machine learning with convolutional neural networks, specific morphological patterns of the tumor, their exact proportion in the tumor tissue, the exact proportion of nucleated tumor cells in the tissue or a region of interest, and much more could be determined. However, there are currently no such decision support systems available for routine use. It has to be further investigated how these systems can improve the work of a pathologist and furthermore support clinicians in standardization, accuracy, and time- and cost-efficiency.

Although no tumor was classified as a multiple-classifier, one tumor (no. 35) showed a subclonal TP53 mutation in less than 10% of tumor cells. As this was a subclonal mutation, this tumor was not classified as a multiple-classifier but gained attention during analysis. Multiple classification has been described and discussed before [8,14], and recent data suggest that tumors classified to the MSI group or to the POLEmut group which harbor a mutation of TP53 should be classified as MSI and POLEmut, respectively [35].

Another very interesting topic is the socioeconomic impact of molecular profiling. Molecular profiling causes additional diagnostic costs, but the potential reduction in treatment cost would presumably outweigh the costs for molecular profiling. However, a detailed socioeconomic analysis was beyond the scope of the current manuscript.

To the best of our knowledge, this is the first report on real-world data relating to molecular profiling in endometrial cancer patients treated at a European university cancer center. Although this study is flawed due to its retrospective design and has limitations in terms of a small case number, we think that these data are clinically sound and relevant. We show that molecular profiling is feasible in everyday clinical life. Molecular profiling did indeed prolong pathologic examination and hence would delay the start of adjuvant therapy by roughly 2–3 weeks; however, this delay may be considered negligible in most cases. Of note, caution has to be exercised with clinical algorithms, including obtaining patients’ informed consent and correct pathologic request statements, as this organizational effort led to a clinical drop-out rate of 16.6% in the current analysis. 

Finally, as shown in this analysis, there are substantial discrepancies between the current ESMO classification system and the integrated molecular risk classification. These discrepancies led to vastly different recommendations for adjuvant therapy in a significant proportion of patients. While the molecular risk classification is on the verge of integration into the daily routine, data from prospective randomized clinical trials are still lacking.

## Figures and Tables

**Figure 1 jpm-11-00048-f001:**
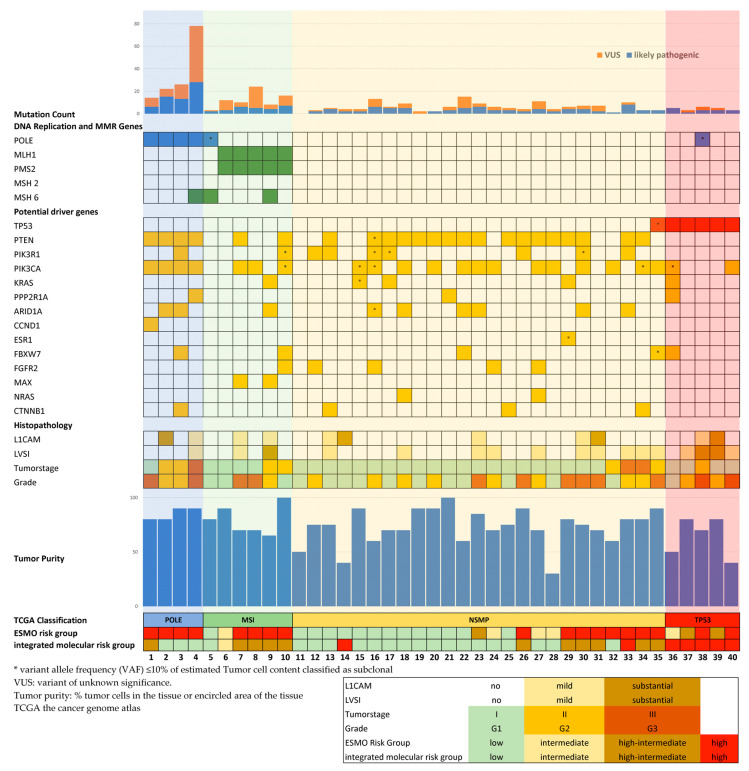
Molecular and histopathologic profiles of 40 patients with endometrioid adenocarcinoma of the endometrium.

**Figure 2 jpm-11-00048-f002:**
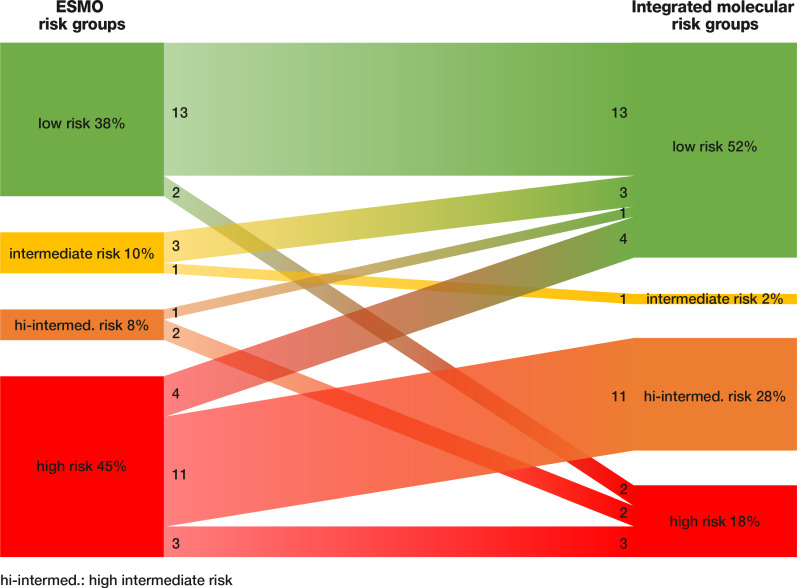
Shifts of patients between risk groups of two different stratification systems.

**Table 1 jpm-11-00048-t001:** Antibodies used for immunohistochemical staining of MMR-proteins and L1CAM.

Antigen	Clone/Clonality	Dilution of Primary Antibody	Company
MLH1	M1/mouse monoclonal	Ready to use	Ventana Medical Systems
PMS2	EPR3947/rabbit monoclonal	Ready to use	Cell Marque (Rocklin, CA, USA)
MSH2	G219-1129/mouse monoclonal	Ready to use	Cell Marque
MSH6	44/mouse monoclonal	Ready to use	Cell Marque
L1CAM	14.10/mouse monoclonal	1:100	BioLegend (San Diego, CA, USA)

**Table 2 jpm-11-00048-t002:** Patient demographics.

Variable	No. (%)/Median (Range)
No. of patients analyzed	40
Patient age	62.1 (29.9–83.6)
BMI	28.5 (18.3–46.0)
FIGO Stage	
Ia	18 (45.0)
Ib	12 (30.0)
II	7 (17.5)
IIIc1	2 (5.9)
IIIc2	1 (2.5)
Nodal involvement	
Nx	4 (10.0)
N0	29 (72.5))
N1	3 (7.5)
N0 (i+)	4 (10.0)
Tumor grading	
G1	15 (37.5)
G2	13 (32.5)
G3	12 (30.0)
LVSI	
No LVSI	23 (57.5)
Mild LVSI	14 (35.0)
Substantial LVSI	3 (7.5)

BMI body mass index, FIGO International Federation of Gynecology and Obestrics, N0 (i+): isolated tumor cells were found in lymph nodes during ultrastaging. LVSI: lymphovascular space invasion.

**Table 3 jpm-11-00048-t003:** Comparison of ESMO risk groups with molecular subtypes and integrated molecular risk groups.

ESMO Risk Group	Molecular Subtypes				Integrated Molecular Risk Group			
	POLE	MSI	NSMP	TP53	Low	Intermediate	High–Intermediate	High
Low	0 (0.0)	1 (6.7)	13 (86.7)	1 (6.7)	13 (86.7)	0 (0.0)	0 (0.0)	2 (13.3)
Intermediate	0 (0.0)	1 (25.0)	3 (75.0)	0 (0.0)	3 (75.0)	1 (25.0)	0 (0.0)	0 (0.0)
High–intermediate	0 (0.0)	0 (0.0)	1 (33.3)	2 (66.6)	1 (33.3)	0 (0.0)	0 (0.0)	2 (66.7)
High	4 (22.2)	4 (22.2)	8 (44.4)	2 (11.1)	4 (22.2)	0 (0.0)	11 (61.1)	3 (16.7)

Chi-squared test: ESMO/molecular subtypes: *p* = 0.038. Chi-squared test: ESMO/integrated molecular risk group: *p* < 0.001. MSI: microsatellite instable group; NSMP: ‘no specific molecular profile’.

## Data Availability

The data presented in this study are available on request from the corresponding author.

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
