# Peer review of "Risk Reclassification of Patients with Endometrial Cancer Based on Tumor Molecular Profiling: First Real World Data"

_jpm, 2021, doi:10.3390/jpm11010048_

Round 1

Reviewer 1 Report

The article entitled “Clinical implications of molecular profiling in patients with endometrial cancer: first real world data” by Oberndorfer et al. investigates the clinical feasibility of molecular profiling and its potential impact on treatment decisions.

In fact, by “molecular profiling”, the authors refer to genetic alteration, histological examination and clinical parameters. The treatment decision is challenged retrospectively. Therefore, I suggest to adapt the title accordingly, and to mention the veritable result: the risk reclassification.

The article is clinically oriented and easy to read. The aims are clear. Strategy is adapted, but some methodological aspects are irregular. The results are interesting, although poorly described. The discussion deserves important revision and limitations must be added.

I have a few specific comments (four major comments) listed below.

Major comment 1:

Robust, standardized and validated IHC protocol and associated quantification are mandatory for such study. The histology method is poorly described, and some irregularities are noted.

First, IHC was performed on 2-µm-thin sections. Section thickness can affect quality and intensity of staining. In general, Ventana recommends that specimens be cut at 4 microns for staining (validation should be performed on duplicate sections at 2, 3, 4, 5, 6, and 7 µm). In particular for MLH1 antibody, manufacturer recommendation is 4 µm. Please, justify why not following the recommendation. Second, IHC is typically performed only once as part of diagnosis procedure; however, replicable staining is not always reached with autostainers (even with clinical grade device), generally due to technical errors. Controls and representative IHC must be illustrated to be conclusive. Third, it is not stated whether examination by pathologists was done in blind. The authors state to have analyzed “a regular nuclear” signal. How many fields of view were analyzed, and at which magnification?  Finally, “tumor stage” and “grade” are interpreted in a semi-quantitative manner in the histolopathology section (illustrated in figure 1). Legend is given, but the method is not provided. Further in the texte, tumor stage seems to have been evaluated by medical imaging. Please, clarify.  

Major comment 2:

Methodological difference and criteria used for risk group classification should be evidenced. If this article aims to sensitize the community to molecular integration for accurate diagnosis/prognosis, the authors should provide a comprehensive comparison, in the method section (lines 155-161) or in the result section. The authors should also specify the most powerful parameters (with a confidence interval if relevant).

Major comment 3:

Can you justify how the data from patient 5 and patient 35 were interpreted and how these patients were decisively assigned to a molecular group? Multiple classification is briefly discussed line 277-282, and refuted very shortly for patient 35 (as explained by the authors a result of subclonal mutation). However, it is not stated from which data tumor purity is calculated, and therefore how this percentage was obtained. Figure 1 is unclear on the purity (15% contamination by TP35 clones should be clarified).

Major comment 4:

Figure 1; Results of patient 11 seem incomplete. If not, then these results are not discussed.

Minor comments:

Abstract

In my opinion, the abstract should mention the retrospective and monocentric approach. The methods used for the profiling should be also given here.

Abbreviation should be omitted here. Therefore, I suggest to revise the sentence “POLEmut, microsatellite  instability,  no  specific  molecular  profile,  and  TP53  mutation respectively”.

For clarity, I recommend to give the number of patients as follows: included, rejected (%), non-characterized and characterized.

Introduction

Line 38: for contextualization, please specify the two historical subtypes.

Results

The results are very short, and do not sufficiently describe the presented data. Please, give more details on the genetic alteration, and briefly justify the classification for patient 5 and patient 35.

If possible, please add in Figure 1 the “current EMO risk group” on top for each patient and the “revised, integrated molecular risk group” below. (Figure 2 would still better reflect the shifts in classification, but it does not indicate the molecular criteria).

Data of figure 2 are presented in the discussion (lines 229-236 should belong to result section).

Discussion

Lines 241-246 : The risk shift, a key outcome of this study, is discussed shortly. The benefit of reclassification is evaluated retrospectively, which allows the authors to fictional medicine. In this exercise, the authors could add socio-economical parameters to fully illustrate the personalized medicine. I suggest the authors to discuss the cost/benefit ratio in terms of economical burden, stress condition, side effects, …

Line 258: Please, briefly revised the sentence to clarify whether the “high response rates for immunotherapy” expected in these 6 patients is speculative or assertive. The given references are for colorectal cancer, with significant differences.

Line 266: In addition to analysis of genetics, the time dedicated to pathological examination by humans is precious and limited. With the onset of slide scanners and digital analysis, the authors should comment the perspectives (such as image analysis by structured deep learning) in terms of data accuracy, robustness and (perhaps) time-saving. Additional patterns could also be taken in account as part of prognosis (immune-infiltration, tumor-associated fibrobalsts…)

The risk of mis-classification is not discussed. Genetics and histology have their intrinsic limits, and data interpretation is (by definition) subject to interpretation. Validity of the method should be discussed, and limitations added.

Writing and figures

Line38, 68, 88, 92, 152, 165, 249, 260, 263,…: missing space.

Line 70, 82: double space

Line 92 replace “where” by ”when”

Line 139: dot missing

Line 163: Missing characters

Line 204-205: “Chi-squared”

Line 285 “this data is” by “these data are”

Figure 1: This figure deserves significant graphical improvement. Legend is hidden. Please revise the incorrect spaces.

Figure 2: Because of the didactic potential of this figure, I strongly recommend the authors to ask the help of a graphical expert for improvement. Please, underline here the re-attribution of 23 patients out of 40.

Reviewer 2 Report

This is a very interesting retrospective study on the differences between molecularly profiled endometrial cancer and the ESMO classification system.

It may serve as a base to initiate prospective-randomised trials, according to the molecular profile of EC.

Thus, I recommend the acceptance of the manuscript.

Author Response

The authors would like to thank the reviewer for his positive review of the manuscript.

Minor spelling mistakes were corrected, as recommended.

Round 2

Reviewer 1 Report

I wish to thank the authors for the accuracy and the precision of their answers. All my comments have been satisfactorily addressed, elegantly discussed, and the revised manuscript was amended accordingly. I particularly appreciate the improvement in scientific precision and the graphic refinement: I have no doubt that this will help the visibility of this article. Technically, the addition of S1 Figures underlines the quality of the conducted research. Scientifically, the reclassification proposed in this study opens significant perspective to improve EC management. For these reasons, I recommend acceptance of this manuscript, although some very specific (technical) details may still be improved; see below.

Extra comments (under appreciation of the Editor):

1) Lines 149 and 150: please, mention the temperature for HIER and for the primary antibody. It may be worth to mention also the parameters for the secondary antibody.

2) There is mention of "isotype-specific antibody (Negative Control (Monoclonal) – Ventana) to preclude unspecific binding" in legend of S1 Fig3. However, line 153 indicates that NC were obtained by omission. Could you clarify whether NC was obtained using an irrelevant antibody (ie. "universal negative control") or by omission?

3) S1 Figures 1 & 2: Labeling is clean and convincing. However, technically, negative controls cannot be obtained only by interpretation of the staining within the same tissue (wrong dilution may cause artefact). In the future, please prefer illustrating both your positive tissue control and negative technical controls.

4) Line 376 can be deleted.
